# Tensile and Fixed Elongation Properties of Polymer-Based Cement Flexible Composite under Water/Corrosive Solution Environment

**DOI:** 10.3390/ma13092155

**Published:** 2020-05-07

**Authors:** Er-Lei Bai, Gao-Jie Liu, Jin-Yu Xu, Bing-Lin Leng

**Affiliations:** 1Department of Airfield and Building Engineering, Air Force Engineering University, Xi’an 710038, China; liusiremail@163.com (G.-J.L.); eihuiyi@163.com (J.-Y.X.); 2College of Mechanics and Civil Architecture, Northwest Polytechnic University, Xi’an 710072, China; 3Air Force Support Department, the PLA Southern Theater Command, Guangzhou 510000, China; 17749132826@163.com

**Keywords:** polymer cement composite, joint sealant, qater immersion, fixed elongation, tensile properties, durability

## Abstract

This study examined the tensile and fixed elongation properties of flexible composite made of styrene–acrylic, vinyl acetate-ethylene copolymer emulsion (VAE emulsion), and cement as cementitious material for airport pavement joint sealant. Quantitative analysis of the elastic recovery ratio and a series of specimen tensile indicators after water immersion, drying–wetting cycles, and corrosive solution (H_2_SO_4_, NaOH, and jet fuel) immersion were performed. Results showed excellent polymer-based cement flexible composite (PCFC) resistance against water and corrosive solution erosion, such as failure mode, elastic recovery, tensile strength, and energy absorption. When the level of water/corrosive solution erosion (immersion time, cycles) were increased, the tensile and fixed elongation properties progressively decreased. Specimens retained more than 60% elastic recovery ratio after water/corrosive solution erosion immersion for 30 days. According to erosion testing as per immersion time in corrosive solution, jet fuel had the maximum effect, NaOH solution had the least effect, and H_2_SO_4_ solution had an intermediary effect. At immersion time in the range of 1–30 days, the tensile strength does not change by more than 0.07 MPa. Within the limits of the fixed elongation tests, cohesive failure occurred after jet fuel immersion for 30 days, adhesive failure occurred after H_2_SO_4_ solution immersion for 30 days but was normal in other cases.

## 1. Introduction

Joints are usually applied in designing and constructing airport pavement concrete to avoid cracks due to temperature and humidity fluctuations [1]. Joint sealant is used to caulk these joints and its quality directly affects the performance and service life of the airport pavement structure [2,3]. Complex environmental conditions such as rain, snow, acid rain, jet fuel leak, and snow-melting agents applied during winter in normal servicing processes affect the performance of joint sealant [4,5,6]. Partly, these joint sealant degradations can be attributed to the negative effects of water/corrosive solutions erosion [7,8]. Ultimately, these destructive reactions such as aging, corrosion, and degradation accelerate joint sealant performance degeneration. Consequently, common joints failures—such as fracture (cohesive failure), debonding (adhesive failure), protrusion, etc.—occur [9,10]. After several joint failures, the durability of the concrete structures is significantly reduced due to the damaged joint sealant adversely affecting takeoff and the plane safety [2,6,11]. Therefore, it is important to conduct research on new materials that are more durable than conventional joint sealants (polyurethane, polysulfide, silicone, etc.) for extended service life of pavement structures [12,13,14].

Polymer-based cement flexible composite (PCFC) is a high-performance composite composed of polymer, cement, inorganic fillers, and admixtures. Cement has high compressive strength, high durability, and is inexpensive. In addition, polymer is highly water-resistant, flexible, and stretchable against joint sealant bending and stretching [15,16,17]. These characteristics have made polymer–cement hybrids the main cementitious composite matrix material with high adhesion performance [18]. Therefore, PCFC is a potentially preferred material to caulk pavement joints due to such benefits [19]. Compared to conventional organic joint sealants, PCFC guarantees long-term service with high performance under complex environmental conditions with extensive prospective application. Many researchers have investigated polymer cement composites for high performance and many studies focus on modification mechanism [20,21,22,23,24], mechanical properties [25,26], engineering application [27,28], etc. Hussain et al. [22] assessed the mechanical performance of polyurethane–cement composites. The results revealed that the new material has high bending tensile strength, bonding strength, and excellent bonding and adhesive property with concrete. The research indicated that the corrosion and frost resistance was substantially increased and surface roughness was decreased. Barnat et al. [24] studied the effect of polysiloxanes on roughness and durability of basalt fiber–reinforced cement mortar. Won et al. [27] developed a kind of high-strength polymer cementitious composites used to protect concrete structures against fire and it shows qualities such as high strength, fireproof, and excellent adhesive property. However, there are still few studies on durability of PCFC as a pavement joint sealant.

In this study, based on optimized material selection and mix ratio design as per our previous studies [29], water and corrosive solution erosion effects on the durability of PCFC analysis was performed. This paper mainly aims to explore the tensile performances of the FPCC material using a vinyl acetate–ethylene copolymer emulsion (VAE emulsion) and Portland cement as the major raw materials. This study aims to explore the durability of PCFC and may provide a reference for the application of PCFC in airport pavement.

## 2. Experimental Details

### 2.1. Materials and Specimen Preparation

The raw materials used to prepare PCFC included [29]: styrene–acrylic and VAE emulsion, the chemical compositions and physical properties of which are listed in Table 1; 42.5R ordinary Portland cement from Lantian Yaobai (Xi’an, China); talcum powder(60% SiO_2_, 30% MgO, 600 mesh) produced by Wantong Powder factory (Dashiqiao, China); heavy calcium carbonate (500 mesh, purity > 99%) produced by Weina Powder factory (Chengdu, China). In addition, some conventional additives were added, such as defoamers (purity: 99%, pH: 5.1, flash point: 174 °C) from San Nopco company (Shanghai, China), dispersants (solid content: 42.5%, purity: 99%, pH: 7.5) from San Nopco company, film-forming additive (C_12_H_24_O_3_ ≥ 99.0%) from Tianyin Chemical industry (Yixing, China), plasticizers (purity: 99%, density: 0.985 g/mL) from Zhiyuan Chemical company (Yixing, China), and silane coupling agents (purity:98%, refractive index: 1.487) from Yingchu Chemical company (Jinan, China). The defoamers reduces the bubbles generated by stirring and various surface-active substances. The dispersants promote the depolymerization and dispersion of powder particles in polymer emulsion. The film-forming additive enhances the plastic flow of the polymer particles and promotes their coalescence to form a film. The plasticizers reduce the vitrification temperature of the polymer and improve its flexibility. The silane coupling agents improve their bonding, water resistance and weather resistance of the mixture. The corrosive media included sulfuric acid (H_2_SO_4_) solution (pH = 1, 0.05 mol/L), sodium hydroxide (NaOH) solution (pH = 13, 0.1 mol/L) and jet fuel no. 3. The ratio of PCFC mixture is shown in the Table 2.

As described in our previous research [29], the procedure for preparing PCFC specimens for fixed elongation and tensile test was as follows: (1) the dispersant, film-forming additive, silane coupling agents, plasticizers and half of the defoamer were added to the mixed emulsion with styrene–acrylic and VAE and homogenized with an electric mixer at 300 r/min stirring rate for 150 s. (2) The cement, talcum powder, and heavy calcium carbonate were mixed and stirred for 3–5 min and these powder materials were then added into the emulsion and homogenized at 700 r/min stirring rate for 10 min. (3) The remaining half of the defoamer was added and stirred at 120 r/min for 3 min and then manually mixed for 10 min to remove the entrained air. (4) The mixture was cast into a hollow volume (50 mm × 12 mm × 12 mm) made up of mortar substrates and blocks of the Chinese standard [30] dimensions. (6) After 28 days of standard curing (*T* = 20 + 2 °C, relative humidity RH > 95%), the PCFC specimen was prepared as shown in Figure 1. The PCFC (50 mm × 12 mm × 12 mm) is cemented to the two mortar substrates (75 mm × 12 mm × 25 mm). The procedures of the test and the size of the specimens follow the Chinese standard [30]. A total of 189 PCFC specimens were prepared.

### 2.2. Experimental Method

#### 2.2.1. Durability Test Conditions and Test Scheme

Five simulated environmental factors pretreatments comprising of water immersion, drying–wetting cycle, and corrosive solution (H_2_SO_4_, NaOH, and jet fuel) immersion for 0–30 days (or 0–20 cycles) were performed to simulate different joint sealant service environmental conditions, such as rain (snow), air-drying, acid rain, jet fuel leakage, etc. The elastic recovery properties from erosion damages were evaluated using fixed elongation tests, and specimen’s appearance or failure mode observation. Moreover, the PCFC specimens’ tensile strength, deformation property, and energy absorption capacity after erosion were compared and analyzed via tensile tests.

All prepared PCFC specimens were divided into control group and durability group for comparison. The PCFC specimens (25 °C) without any durability test condition were used as the control group. The durability group of PCFC specimens was subjected to different durability test conditions such as: water immersion, drying–wetting cycle, corrosive solution immersion (H_2_SO_4_, NaOH, and jet fuel). Apart from different durability test conditions, the specimens were not subjected to other differential treatments. The durability group test schemes are listed in Table 3. There are 21 durability test conditions, depending on the environment condition and immersion time (or cycles). Nine specimens were allocated to each durability test condition, three for fixed elongation test, three for tensile test and the remaining three specimens were stand-by for contingency.

#### 2.2.2. Fixed Elongation and Tensile Test

The tests were completed according to our previous studies [29]. The fixed elongation test was conducted using a self-made device (Figure 2). The device and the fixed elongation test follow the Chinese standard (GB/T 13477-2002, Test Methods for Building Sealing Materials). Firstly, the PCFC specimens were elongated with preset width (60% of its original width), and then stretched for 1 day by setting the position blocks. Thereafter, the specimens were placed horizontally without the position blocks for 1 day (Figure 3). Finally, the specimen width after elastic recovery was measured and the elastic recovery ratio (*R*_e_) is calculated via Equation (1).
(1)$$R_{e}=\frac{W_{1}-W_{2}}{W_{1}-W_{0}}\times 100\%$$
where *W*_0_, *W*_1_, and *W*_2_ are the initial width, width after elongation, and width after elastic recovery of the specimen, respectively.

The tensile test was conducted using electronic tensile testing machine (HS-3001B, produced by Shanghai Heson company, City, Country) at 5 mm/min loading rate (Figure 4). The corresponding load–displacement curves were recorded.

## 3. Results and Discussion

### 3.1. Effect of Water

#### 3.1.1. Elastic Recovery Ratio

Figure 5 shows the appearances of PCFC specimens in the fixed elongation test after water immersion for 0–30 days. Clearly, the PCFC surface turned whiter over time and no cohesive failure or adhesion failure occurred in PCFC. Moreover, when the immersion time was prolonged to 30 days, the white color on surface of the specimens deepened and the surface was no longer smooth and flat, but swelling and foaming appeared. This shows that there is significant effect of water immersion on the specimen internal components. In addition, the elastic recovery ratio of PCFC decreased continuously over immersion time as shown in Figure 6. Figure 6 indicates that the longer the immersion time, the greater the decline.

Figure 7 demonstrates the visual appearances of PCFC specimens in the fixed elongation test after drying–wetting cycle for 5–20 cycles. Evidently, no cohesive failure or adhesive failure occurred in the specimens even after 20 cycles and PCFC demonstrated potential fixed extension performance. Next, we examined the effect of drying–wetting cycle on elastic recovery ratio of the specimens. Figure 8 reveals that the PCFC elastic response rate decreased more rapidly with increasing cycles. Notably, compared to the specimen without cycles, the elastic recovery ratio after 20 cycles was less than 60%.

#### 3.1.2. Tensile Properties

Figure 9 illustrates the changes in tensile strength ($f_{t}$), elongation at break ($\delta_{b}$), peak strain ($\varepsilon_{P}$), tensile toughness ($T_{t}$), and pre-peak tensile toughness ($T_{t,b}$) with immersion time in tensile test analysis. Results indicate: (1) A tendency towards lower $f_{t}$ in the specimens with increasing immersion time. The tensile strength increase trend is 7.8%, 20%, 34%, and 45.3% in water immersion for 1, 7, 15, and 30 days, respectively (Figure 9a). (2) There is a significant negative correlation between $\delta_{b}$ and the immersion time, with $\delta_{b}$ loss up to 28%. In contrast, $\varepsilon_{P}$ of the specimens displays a rapid increment rate after water immersion for 1 d and the increasing has continued since then but has slowed (Figure 9b). (3) Growth declines in the specimens $T_{t}$ with a significant correlation to increasing immersion time. A similar trend on $T_{t,b}$ is observed (Figure 9c).

Tensile properties of PCFC specimens after drying–wetting cycle are shown in Figure 10a. The additional cycles lead to slight rise in tensile strength at first and significant decline followed. Raising the cycles caused elongation of the specimens at a break extremely weaker against tensile loading, hence an inclination towards lower residual elongation at break is observed in Figure 10b. Likewise, a similar trend is observed in peak strain, tensile toughness, and pre-peak tensile toughness of the specimens (Figure 10b,c). Results imply that the PCFC specimens experienced reduction in $\delta_{b}$, $\varepsilon_{P}$, $T_{t}$, and $T_{t,b}$ of 30%, 36%, 38%, and 39% from 20 cycles, respectively.

#### 3.1.3. Discussion

In line with previous studies [31], our data also show that water immersion has a significant negative correlation with elastic recovery and tensile properties among specimens. The degradation mechanism can be considered from two dimensions, the adhesive interface and PCFC interior degradation. The adhesive interface degradation: a synergistic effect is caused by the strong polarity of water molecules and numerous capillary pore structures at the adhesive interface. Therefore, water molecules quickly penetrate the adhesive interface to form a weak interface layer of water molecules [32]. Furthermore, chemical bonds and intermolecular forces at adhesive interface are destroyed by extended immersion time, and the adhesion between PCFC and mortar substrates drops. This is supported by the change from cohesive failure to adhesive failure in specimens immersed for 30 days.

With cohesive interface degradation: water immersion has three negative effects on the interior of the specimens, plasticizing, hydrolysis and softening with swelling. (1) Plasticizing—water molecules easily formed hydrogen bonds with polar groups in polymer macromolecules [33]. Meanwhile, hydrogen bonds and other secondary bonds between polymers molecules are broken. Therefore, intermolecular forces are weakened, and the distance between molecular chains increases. This trend leads to polymer plasticization and reduces material modulus. (2) Hydrolysis—due to chemical reactions between hydrolyzable groups in polymer macromolecules (such as ester, carboxyl, hydroxyl, etc.) and water molecules, polymer macromolecular chains are hydrolyzed and broken, deteriorating PCFC specimens. (3) Softening with swelling—free volume inside PCFC is the basis for free movement and rotation of polymer molecular chains to change their conformations [34]. Increased water immersion reduces the free volume inside the PCFC. Thus, the PCFC flexibility changes with softening and swelling, which is consistent with previous studies [5]. This is evident from Figure 5. Altogether, these results suggest that the tensile strength, elongation at break, tensile toughness, pre-peak tensile toughness, and elastic recovery ratio decreases after elongation and the peak strain increases. This is mainly because the PCFC polymer molecular chains become softer after immersion, and larger deformation are caused by the small force.

In terms of the drying–wetting cycle, initially (few cycles), the non-hydrated cement inside PCFC reacts with the infiltrated water through secondary hydration. Consequently, the PCFC tensile strength increases slightly, while the elastic recovery rate, deformation, and energy consumption indexes decreases. In addition, water immersion contributes to gradual PCFC growth PCFC deterioration with the addition of cycles [35]. Besides, the adverse effects of water immersion on the properties gradually exceeds the reinforcement effects of cement secondary hydration [36]. Accordingly, up to certain cycles, the tensile and elastic recovery properties clearly decrease. Essentially, secondary hydration reaction of cement also exists in the water immersion test.

### 3.2. Effect of Corrosion Environment

#### 3.2.1. Elastic Recovery Ratio

Figure 11 illustrates the images of PCFC specimens under acid solution, alkaline solution and jet fuel for 1–30 days. The surface of PCFC specimens gradually whitens with immersion time, but no other significant change or damage occurred. For instance, the mortar substrate is highly corroded while the PCFC adhered to it maintaining a good fixed elongation performance. On the other hand, severe cohesive failure and adhesion failure occurred in PCFC specimens under jet fuel environment for 30 days. Similarly, after being immersed in jet fuel for 15 days, PCFC maintained a positive fixed elongation performance. Moreover, from Figure 12, the above phenomenon is confirmed. These data suggest that the elastic recovery ratio of the specimens drops under these three corrosive environments. This downward trend of the elastic recovery ratio of the specimens is specifically rapid under a jet fuel environment. When immersed in jet fuel for 30 days, the broken specimens can no longer recover.

#### 3.2.2. Tensile Properties

Figure 13 depicts the variations in tensile properties of PCFC specimens. The changes in tensile strength, elongation at break, tensile toughness, and pre-peak tensile toughness as shown by the graphs (Figure 13a,b,d,e) are similar. These indexes all decrease with increasing corrosion time. Contrarily, the effects of acid solution and alkaline solution were milder than those of jet fuel. In the entire jet fuel immersion process, the PCFC degeneration was obvious. Additionally, after 30 days of jet fuel corrosion, the $f_{t}$, $\delta_{b}$, $\varepsilon_{P}$, $T_{t}$, and $T_{t,b}$ of the specimen was only 34%, 39%, 36%, 11%, and 11% of the original, respectively. Conversely, the peak strain of PCFC immersed in acid solution or alkaline solution increase, which is similar to that of water immersion. It is possible that the acid and the alkaline solution can react like water to the specimens.

#### 3.2.3. Discussion

Clearly, the three corrosive liquids—H_2_SO_4_, NaOH, and jet fuel—weakened the joint sealant’s elongation adhesive performance and tensile performance to different levels. Jet fuel immersion had the maximum deterioration effect, while acid and alkali solutions slightly weakened the specimens. The degradation mechanism of jet fuel-immersed specimens may be explained by the fact that the jet fuel molecules gradually penetrated into the joint sealant’s interior, interacting with the polymer macromolecular chain [37]. Meanwhile, the van der Waals forces, hydrogen bond, and other bonds of the polymer macromolecular chain were destroyed, causing fast swelling, and gradual softening of the joint sealant interior [8]. As the immersion time increased (the amount of jet fuel infiltration continues to rise), the swelling and softening effect of jet fuel on the joint sealant intensified, causing the internal polymer macromolecular chains to gradually unfold, break, and slip [38]. This resulted in significantly low elongation adhesive performance and tensile performance of the joint sealant. Experimentally, it is evident that the joint sealant immersion in jet fuel for 30 days lost its elastic recovery ability, and the tensile strength, deformation, and energy consumption indicators have been greatly reduced. In addition, as the jet fuel molecules are concentrated at the bonding interface between the joint sealant and the cement mortar substrate, and forms the jet fuel membrane, the adhesion between the joint compound and the substrate is reduced. This is manifested in tensile tests, where the failure mode of the joint sealant changes from cohesive failure to adhesive failure as the jet fuel immersion time increases.

After H_2_SO_4_/NaOH solution immersions, the joint sealants still maintained good elongation adhesive performance. Similarly to the water immersion test outcomes, the mechanical properties and elastic recovery rate gradually decreased with immersion time. The is because the polymer has good resistance to acid and alkali [39,40]. In other words, H_2_SO_4_/NaOH solution has similar erosional effect to PCFC as water, which is consistent with previous studies [41]. These factors explain the relatively good correlation between H_2_SO_4_/NaOH solution immersion test and water immersion test. Besides, it is worth noting that the effect of jet fuel immersion on the performance of joint sealant is significantly higher than the effect of water immersion.

### 3.3. Durability Analysis

The performance comparison of the PCFC and other conventional joint sealants is important. Actually, the PCFC was applied to two Chinese airports (a military airport in Tibet, another one in Guangdong province) and observed for two years to test the actual performance. The brief application process of the PCFC is shown in the following Figure 14 and the appearance of the PCFC after 28 days of curing is shown in Figure 15. Figure 16 shows the PCFC after 2 years of service. It can be seen that cohesive failure or adhesive failure did not occur in the PCFC of the two airports, which has firm adhesion with the pavement and shows good sealing performance. In addition, the appearance of conventional joint sealants after one year is shown in Figure 17. It is easy to find that serious cohesive failure and adhesive failure have occurred. In summary, the durability of PCFC is significantly better than the conventional joint sealants in actual service.

Based on 12 current standards about sealing material for concrete pavement and construction joints, Wang proposed the performance requirements of durability for joint sealants of airport pavement [42]. The durability requirements of the joint sealants proposed by Wang related to the test in this paper and some test results of the PCFC are listed in Table 4. It is easy to see that the performance of PCFC greatly exceeds the requirements.

## 4. Conclusions

This study set out to investigate the effect of water immersion, drying–wetting cycle, corrosive solution (H_2_SO_4_, NaOH, and jet fuel) immersion on the tensile and elastic recovery properties of the polymer-based cement flexible composite for airport pavement joint sealant. The sealants are made up of styrene–acrylic and VAE emulsion and were analyzed via tensile and fixed elongation tests. The main findings from this study can be summarized as follows:PCFC exhibited high erosion resistance against water and corrosion solution. Within fixed elongation experimental limits, PCFC specimens had good cohesive performance and retained more than 60% elastic recovery ratio.After water immersion, the joint sealant became soft under the ‘plasticizing’ action of water. The tensile strength decreased by at most 45% and the peak strain increased by at most 34%. After the drying–wetting cycle treatment, the tensile properties of the joint sealant decreased significantly except for the tensile strength which initially increased and thereafter decreased with increasing cycles.After acid solution and alkali solution immersion, the overall decline of the tensile, deformation, energy consumption indexes and elastic recovery rate of the PCFC was minor and no significant degradation performance occurred. with the extension of the immersion time. At immersion time in the range 1–30 days, the tensile strength does not change by more than 0.07 MPa. After jet fuel immersion, the tensile properties of the joint filler significantly deteriorated.

The experimental results showed that polymer-based cement flexible composite is highly durable against water/corrosive solution erosion. However, this study only covers limited simulated environmental factors. Although these results shed some light on the performance of PCFC joint sealant material, further studies on the durability properties of PCFC in a wide service environment range are deemed necessary.

## Figures and Tables

**Figure 1 materials-13-02155-f001:**
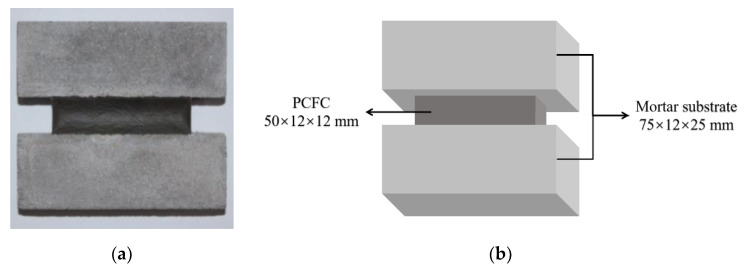
(**a**) The PCFC specimen for fixed elongation and tensile test. (**b**) Schematic diagram of the PCFC specimen.

**Figure 2 materials-13-02155-f002:**
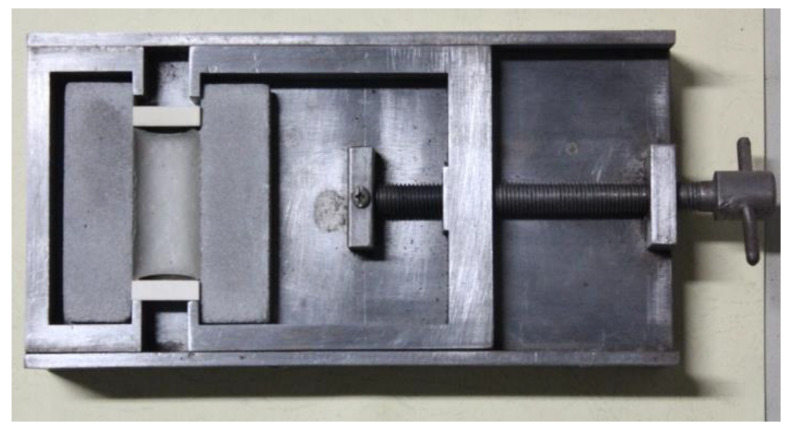
Fixed elongation test set up.

**Figure 3 materials-13-02155-f003:**
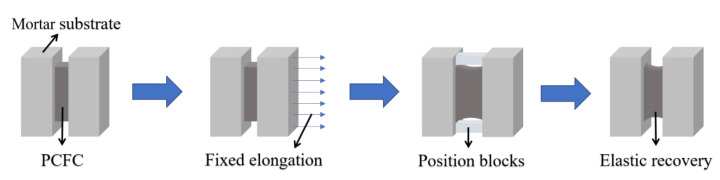
Fixed elongation test process.

**Figure 4 materials-13-02155-f004:**
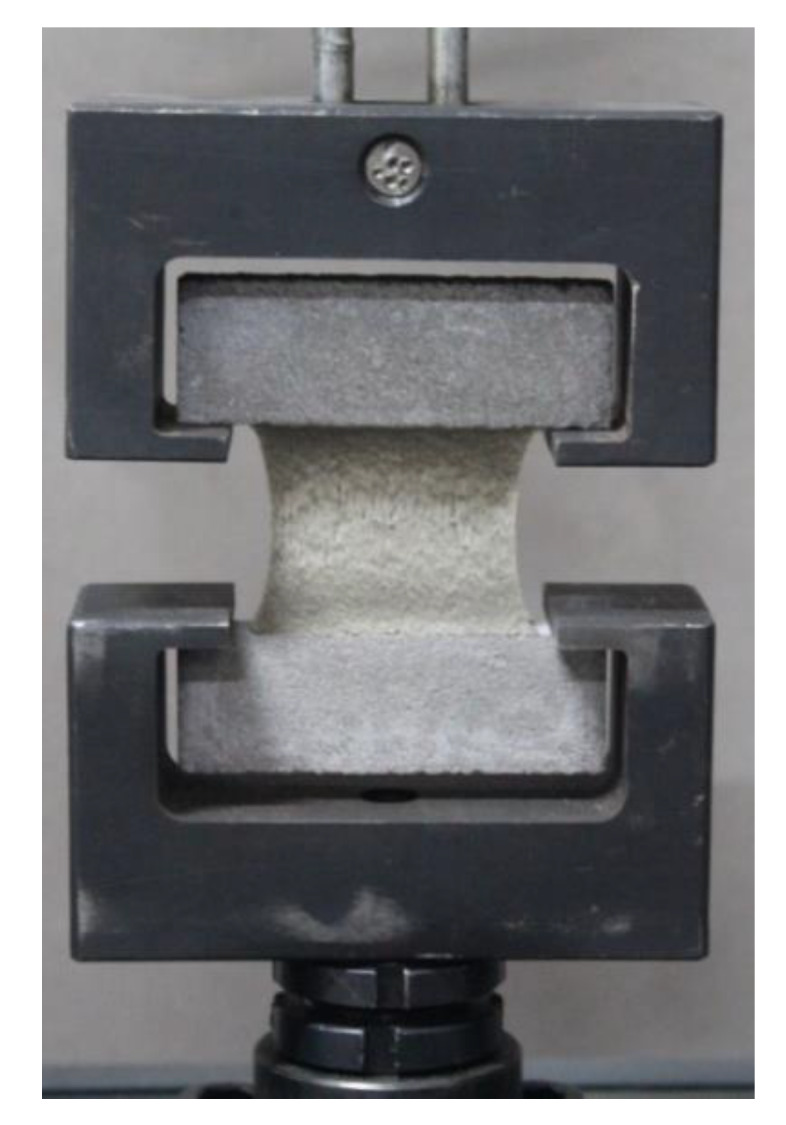
Tensile test set up.

**Figure 5 materials-13-02155-f005:**
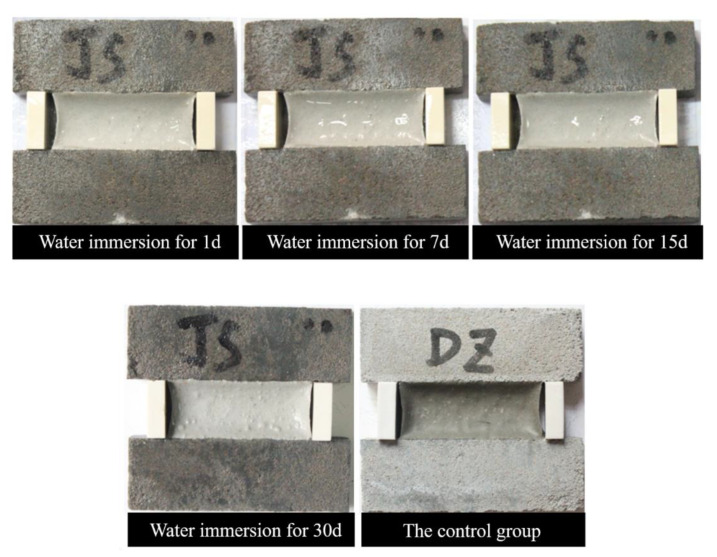
Fixed elongation morphology characteristics of PCFC under different water immersion time.

**Figure 6 materials-13-02155-f006:**
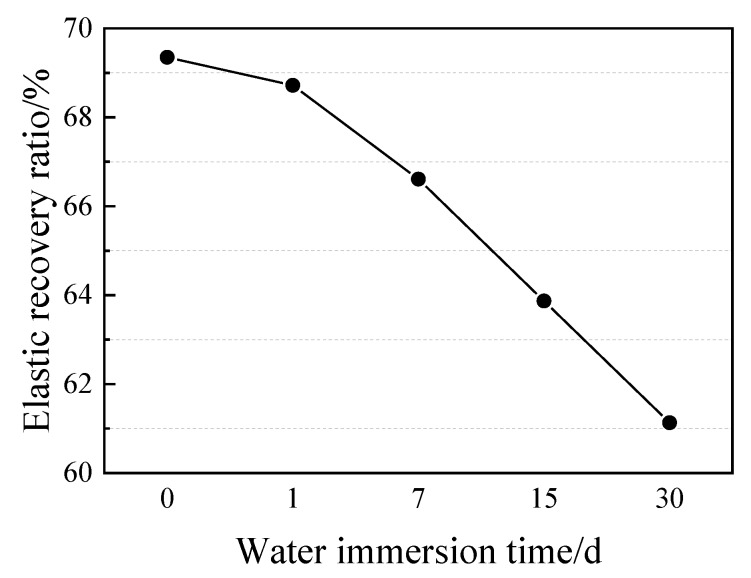
Elastic recovery ratio of PCFC under different water immersion time.

**Figure 7 materials-13-02155-f007:**
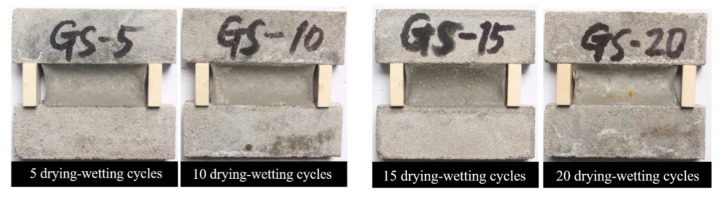
Fixed elongation morphology characteristics of PCFC under different drying–wetting cycles.

**Figure 8 materials-13-02155-f008:**
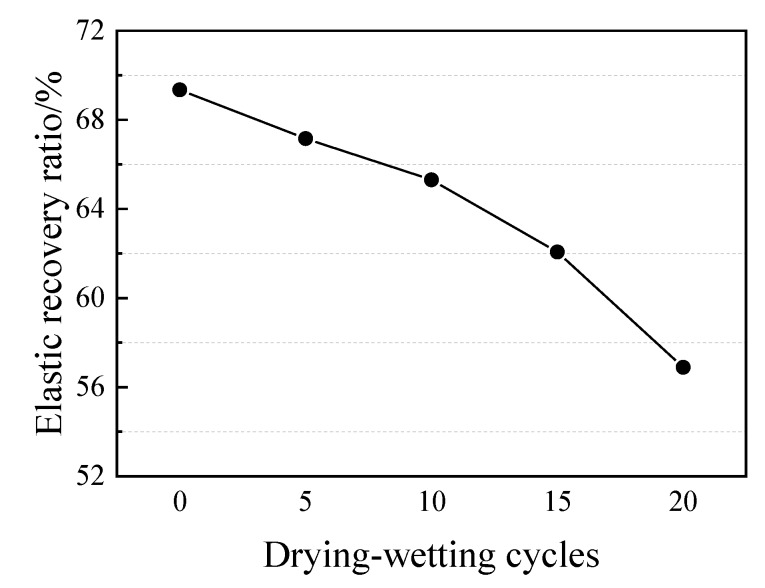
Elastic recovery ratio of PCFC under different drying–wetting cycles.

**Figure 9 materials-13-02155-f009:**
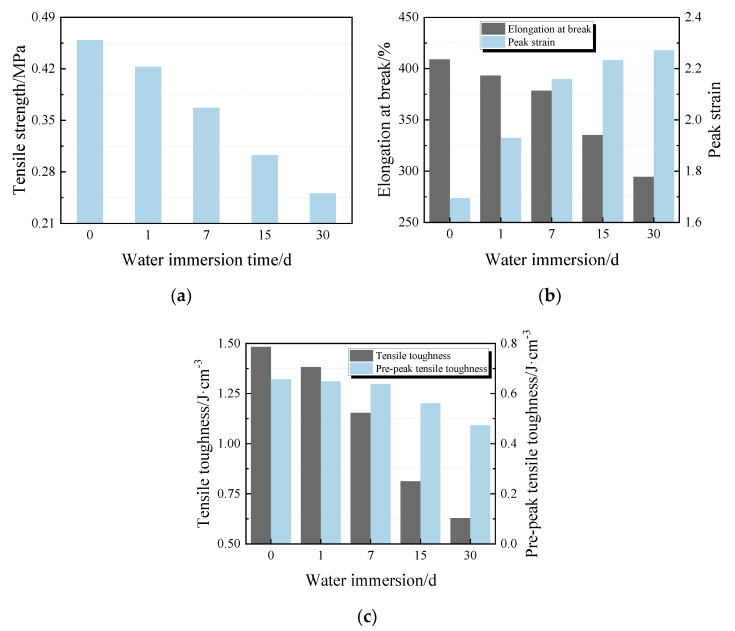
Tensile properties of PCFC under different water immersion time. (**a**) Tensile strength. (**b**) Elongation at break and peak strain. (**c**) Tensile toughness and pre-peak tensile toughness.

**Figure 10 materials-13-02155-f010:**
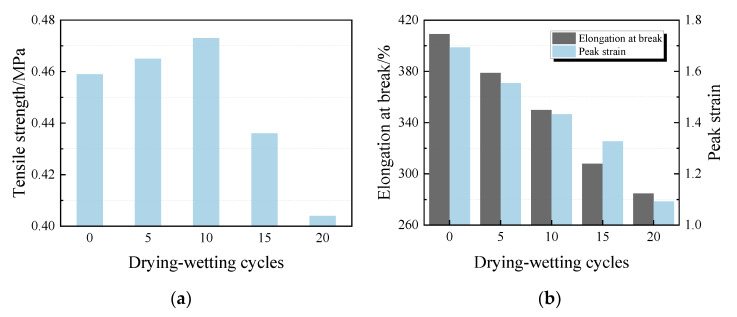
Tensile properties of PCFC under different drying–wetting cycles. (**a**) Tensile strength. (**b**) Elongation at break and peak strain. (**c**) Tensile toughness and pre-peak tensile toughness.

**Figure 11 materials-13-02155-f011:**
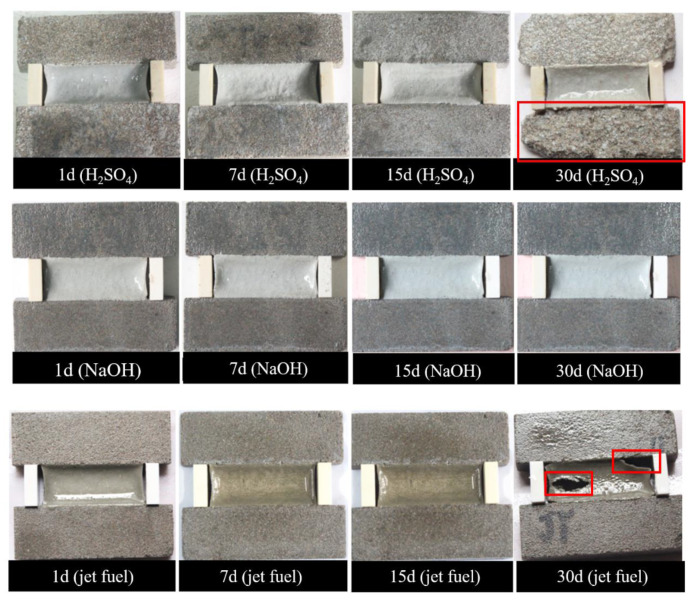
Fixed elongation morphology characteristics of PCFC under different corrosion solution immersion time.

**Figure 12 materials-13-02155-f012:**
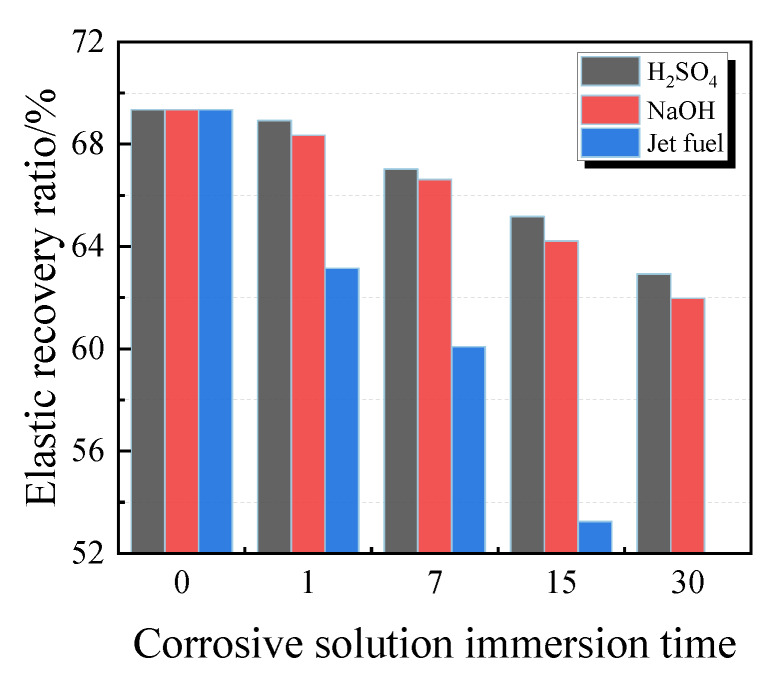
Elastic recovery ratio of PCFC under different corrosion solution immersion time.

**Figure 13 materials-13-02155-f013:**
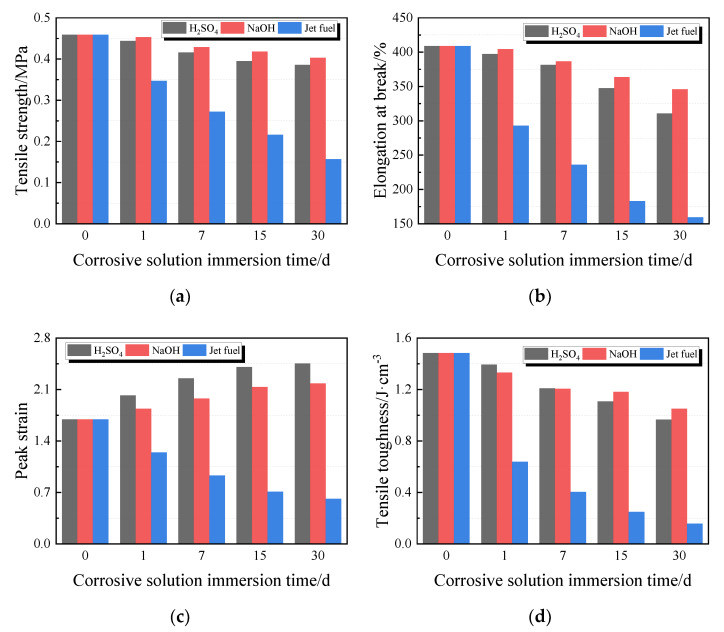
Tensile properties of PCFC under different corrosion solution immersion time. (**a**) Tensile strength. (**b**) Elongation at break. (**c**) Peak strain. (**d**) Tensile toughness. (**e**) Pre-peak tensile toughness.

**Figure 14 materials-13-02155-f014:**
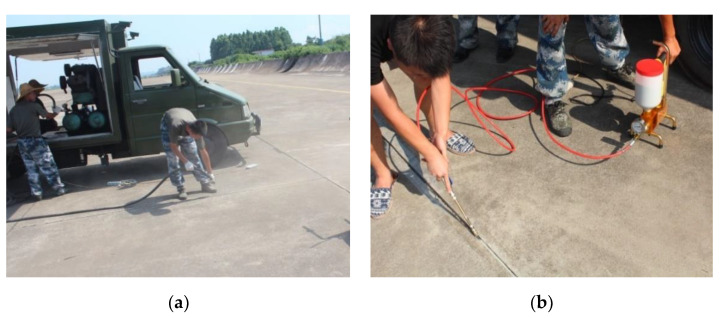
Brief application process. (**a**) Blowing joints by high pressure air gun. (**b**) Injecting the PCFC.

**Figure 15 materials-13-02155-f015:**
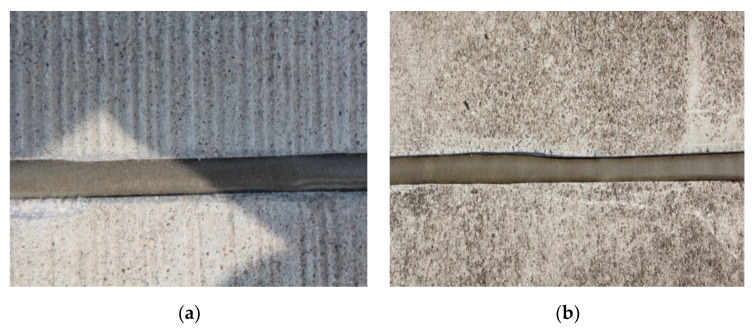
The appearance of the PCFC after 28 days of curing. (**a**) The military airport in Tibet. (**b**) The military airport in Guangdong province.

**Figure 16 materials-13-02155-f016:**
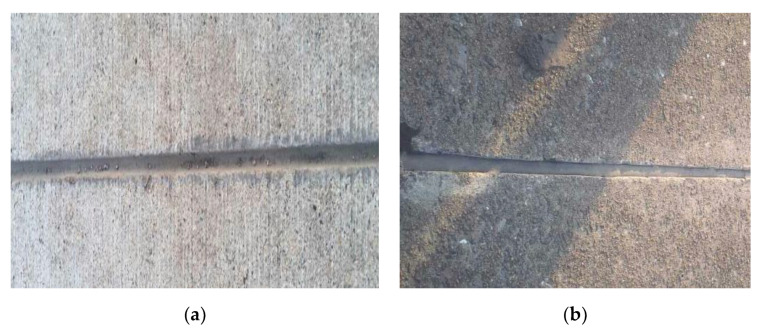
The appearance of the PCFC after two years. (**a**) The military airport in Tibet. (**b**) The military airport in Guangdong province.

**Figure 17 materials-13-02155-f017:**
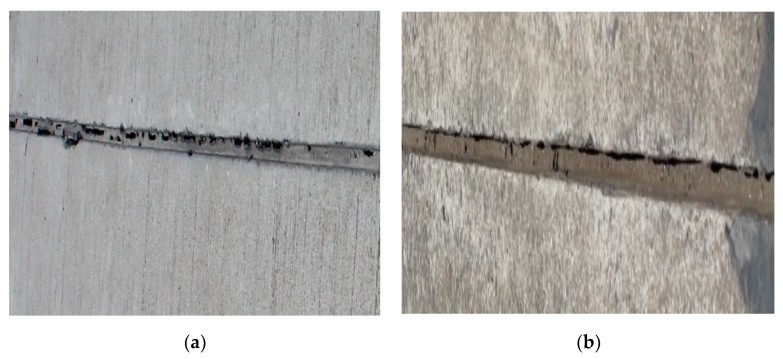
The appearance of the conventional joint sealants after one year. (**a**) The military airport in Tibet. (**b**) The military airport in Guangdong province.

**Table 1 materials-13-02155-t001:** Chemical compositions and physical properties of styrene–acrylic and VAE emulsion

Emulsion	Chemical Compositions	Solid Content (%)	Viscosity (mPa·s)	pH	*T_g_* (°C)	Minimum Film Forming Temperature (°C)	Average Granularity (μm)
Styrene–acrylic	Styrene–acrylate copolymer	56 ± 1	400–1800	7.0–8.5	−7	0	0.1
VAE	Ethylene–vinyl acetate copolymer	55 ± 1	1500–5000	4.5–6.0	−10	0	1.5

**Table 2 materials-13-02155-t002:** Composition ratio of PCFC mixture

Materials	Quantity (kg/m^3^)	Percentage (%)
Styrene–acrylic emulsion	1086.8	43.47
VAE emulsion	585.2	23.41
Cement	234.1	9.36
Talcum powder	217.4	8.70
Heavy calcium carbonate	217.4	8.70
Dispersant	18.7	0.75
Defoamer	11.7	0.47
Film-forming additive	100.3	4.01
Silane coupling agents	11.7	0.47
Plasticizers	16.7	0.67

**Table 3 materials-13-02155-t003:** Durability group test schemes

Environmental Effects	Conditions	Processes
Effect of water	Water immersion	The specimens were immersed into water (maintained at 25 °C) for 1, 7, 15, and 30 days, respectively. Afterwards, the surface moisture was removed and their mechanical properties immediately tested.
	Drying–wetting cycle	The specimens were immersed into water (maintained at 25 °C) for 2 days, and then dried in oven (25 °C) for 1 day (one cycle). The specimens were subjected to 5, 10, 15, 20 cycles and the mechanical properties immediately tested.
Effect of corrosive solution	Acid solution immersion	The specimens were immersed into corrosive solution (H_2_SO_4_, NaOH and jet fuel) for 1, 7, 15, and 30 days. Afterwards, the surface solution was removed and the mechanical property immediately tested.
	Alkaline solution immersion	
	Jet fuel immersion	

**Table 4 materials-13-02155-t004:** Durability test results of the PCFC

Performance	Part of the Durability Requirements Proposed by Wang [42]	Result
Resistance to water	Elongation with 60% of its original width after water immersion (23 °C × 24 h), no failure	No failure, 61.13% elastic recovery (23 °C × 30 days)
Resistance to the jet fuel	Elongation with 60% of its original width after jet fuel immersion (23 °C × 24 h), no failure	No failure, 53.25% elastic recovery (23 °C × 15 days)
Resistance to acid	N/A	No failure, 62.93% elastic recovery (23 °C × 30 days)
Resistance to alkali	N/A	No failure, 61.97% elastic recovery (23 °C × 30 days)

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
