# Peer review of "Tensile and Fixed Elongation Properties of Polymer-Based Cement Flexible Composite under Water/Corrosive Solution Environment"

_materials, 2020, doi:10.3390/ma13092155_

Round 1

Reviewer 1 Report

The manuscript entitled "Tensile and fixed elongation properties of Polymer-based cement flexible composite under water/corrosive solution environment” examined the tensile and fixed elongation properties of polymer-based cement flexible composite (PCFC) for airport pavement joint sealant. Quantitative analysis of the elastic recovery ratio and a series of specimen tensile indicators after water immersion, drying-wetting cycles, and corrosive solution immersion were performed.

This paper summarizes an interesting experimental study. However, there is one concern about this experiment work. To judge the excellent behavior of new material, comparisons with other typical used materials should be carried out. In lines 26-28, the authors confirmed that "PCFC is highly resistance against water/corrosive solution erosion, shedding light on its performance as effective joint sealant material". However, no comparisons with other existing material were obtained. I think the authors need to edit their manuscript to show up the properties of PCFC under different parameters and the judgment on its performance to be excellent or week should be done with comparisons with other used materials under water/corrosive solution environment. 

Therefore, the manuscript lacks clarity and should not be accepted in this current condition. This reviewer recommends major editing and resubmitted for re-review.

Some technical comments:

  • The authors should increase their discussion on previous related research and highlight how their study is providing a different approach or adding significantly to what has been done.
  • The end of the sentence should be after the cited reference. This note is addressed for most of the sentences in the manuscript.
  • Line 37: It should be "affect" not "affects".
  • Line 42: The grammar of this sentence should be improved. “reduces” should be in passive. It should be “is significantly reduced” or “was significantly reduced”.
  • Figure 1: Dimensions of the specimens should be provided. Also, more details should be provided on the figure to define each component of the specimen.
  • Line 98: This expression should be "All prepared PCFC specimens were".
  • Line 98: I recommend using another name for the "experimental group". It does not give a meaning like a name "control group".
  • Line 102: What do you mean by "The other experimental conditions remained constant"?
  • Line 104: What does it mean. I cannot figure out what do you mean by this sentence.
  • Lines 107-108: Which standard does this test follow? What do you mean by " a self-made device"? Results from experiments should follow the current standards in the procedures of the test and the size of the specimens.
  • Line 112: "is" should be added before "calculated".
  • Lines 133-134: The grammar of this sentence should be improved.
  • Line 134: I think you mean "Figure 6" not "Figure 5".
  • Figure 6: I recommend using a curve chart instead of a Bar chart to better illustrate the declination of the elastic recovery ratio. The same note is for Figure 8.

Reviewer 2 Report

Thank you for your manuscript submission. It seems like an interesting topic, however I do believe that this study falls short in communicating what is the current performance standard for joint sealants, and how the proposed PCFC material compares against conventional joint sealants. To answer this question, a comparison should have been made with the proposed PCFC sealant and a conventional sealant under the same strenuous environmental conditions to understand whether this material is an effective substitute.

I also offer the following comments:

-In line 47, Reword 'strong strength' to 'high compressive strength'

-In lines 68-70, the results should be discussed in the results section, not in the introduction.

-Reword line 73, as styrene-acrylic and VAE emulsion are not the only ingredients to prepare PCFC.

-In section 2.1, A discussion of the role or purpose of each additive would be beneficial.

-What does 'rest were reserved' mean in line 104? Please clarify.

-Is the fixed elongation test based on a standardized test? Or a modification to a standardized test?

-What is a wet-dry cycle consisting of? For instance is it 8 hours of immersion and 16 hours of drying? Please clarify.

-Figure 9 should have labels (e.g. Figure 9a, Figure 9b, Figure 9c), and the caption should mention the sub-figures. Same goes for Figure 10, Figure 13,  etc.

-How does the PCFC's performance compare to other conventional joint sealants? What is the current standard of durability?

Reviewer 3 Report

Tensile and Fixed Elongation Properties of Polymer based Cement Flexible Composite under Water/Corrosive Solution Environment

Er-Lei Bai, Gao-Jie Liu, Jin-Yu Xu and Bing-Lin Leng

General Comments:

Thank you for the opportunity to review this manuscript. The paper presents the experimental data on airport pavement joint sealant. The results are generally well presented, quality of the paper is good – authors focus sufficient on the both preparation and editing of the manuscript as well as the formulation of the presented statements. In terms of originality the manuscript suits in the general trend of seeking sealing materials with good mechanical properties and durability, thus, this topic will be interesting for the concrete technologists.

Unfortunately, content have only a moderate interest for the space required, because present only experimental data. The work only solves practical problems and not pretends to broader theoretical summing-up. There is no scientific discussion with other authors' work done. Since these parts are missing, the impression is that the work deals only with local problems. For this reason, some questions concerning this manuscript should be answered by the authors for improving the paper. On the other hand, this work can have practical value.

Specific Comments:

  1. I didn’t like Abstract very much - a lot of declarative phrases, but lacking accurate results. I think it could be rethought and rewritten.
  2. Page 1, line 30. Points are usually written after the literature source, not before it.
  3. The introduction needs to be supplemented. It is not informative enough – it only lists what work has been done, but does not provide any concrete results.
  4. Page 2, lines 60 and 61. The introduction should only contain the latest scientific and technical information, so the sentence "Accordingly,..." is redundant.
  5. Page 2, lines 61–68. This is not the subject of the introduction but of the experimental part and should be moved to it (if not repeated).
  6. Page 2, lines 68–70. This is not part of the introduction, but part of the summary of the work.
  7. Page 2, line 74. Table 1 does not indicate the chemical composition.
  8. Page 2, lines 74–80. This paragraph needs to be rewritten because too little information is provided on the materials used. Unspecified manufacturers, purity of some materials provided and others not; etc.
  9. Page 2, lines 79–80. Concentration of sulfuric acid (H2SO4) solution (pH=1), sodium hydroxide (NaOH) solution (pH=13) and type of jet fuel should be reported.
  10. Page 2, line 82. Is the Quantity (kg/m3) really correct? In addition, the percentage composition of the mixture should be provided.
  11. Table 2. Please explain why you have chosen this composition of the mixture. Maybe changing the ratio of components would give even better results.
  12. Page 3, line 93. How many samples did you form at one time?
  13. Page 3, line 98. After how many samples were in the control and experimental groups?
  14. Page 4, line 108. Please describe the accuracy of self-made device.
  15. Page 4, line 116. Please indicate the manufacturer and brand of the tensile testing machine.
  16. Section 3.1 provides a lot of experimental data, but there is no explanation as to whether the values obtained are appropriate. Based on the technical literature and the work of other authors, this section needs to be expanded and made more result-oriented.
  17. I would like the Conclusion to better reflect the concrete results of the work.

            I think that after a major revision including the suggested changes the paper could be accepted for publication in Materials.

Round 2

Reviewer 1 Report

The authors have addressed all the reviewer's comments in the revised manuscript. Therefore, the manuscript in current format is appropriate to be published in this journal.

Author Response

Dear Reviewer,

We deeply appreciate the time and effort you have spent in reviewing our manuscript entitled “Tensile and Fixed Elongation Properties of Polymer-based Cement Flexible Composite under Water/Corrosive Solution Environment (Manuscript ID: materials-775396). Your comments are all valuable and very helpful for revising and improving our paper. At last, we appreciate for Reviewer’s warm work earnestly, and thank you for the approval of the revision.

Thank you and best regards.

Reviewer 2 Report

Thank you for your manuscript submission. This revision significantly improves the shortcomings of the first version.

I offer the following comments:

The last paragraph of the introduction should not have a summary of the test results. This would be appropriate in the abstract and conclusion, but not in the introduction. Rather, the last paragraph of the introduction should clarify the objectives of this study, and the specific contributions the study's findings will make (i.e. what problem it aims to solve, or what knowledge gap it aims to address).

In Line 311, this statement must be removed: "Sorry we did not reflect the relevant work in the manuscript". Please make sure that this paragraph does not sound like a direct response to the reviewers' comment, but rather a continuation of the manuscript's findings.

Table 1 doesn't really mean anything unless you summarize what these standards characterize as a durable joint sealant material.

Tables 2 and 3 could be merged together. You could have in one column the requirement proposed by Wang, and on the other column your results with PCFC. Table 3 seems to indicate no failure, but doesn't seem to show your actual results (e.g. the actual % elongation of its original width). In addition, Table 2 needs to have Wang cited in the caption.

In the conclusion section, what does "45 34" mean? For context, this was the paragraph: "PCFC exhibited high erosion resistance against water and corrosion solution. Within fixed elongation experimental limits, PCFC specimens had good cohesive performance and retained more than 60% elastic recovery ratio. 45 34"

Reviewer 3 Report

It's nice that the authors took note of the comments so quickly and qualitatively. The authors have done enough to correct the first version of their work. In their response, they explained well what they were focusing on. The authors have largely taken into account my comments and suggestions. They rewrote the Abstract, clearly articulated the conclusions. .Hence, while appreciating that the results are generally well presented, quality of the paper is good – authors focus sufficient on the both preparation and editing of the manuscript as well as the formulation of the presented statements – in my opinion, at this level of readiness, the paper could be accepted for publication in the Materials.

Author Response

Dear Reviewer,

We deeply appreciate the time and effort you have spent in reviewing our manuscript entitled “Tensile and Fixed Elongation Properties of Polymer-based Cement Flexible Composite under Water/Corrosive Solution Environment (Manuscript ID: materials-775396). Your comments are all valuable and very helpful for revising and improving our paper. At last, we appreciate for Reviewer’s warm work earnestly, and thank you for the approval of the revision.

Thank you and best regards.

This manuscript is a resubmission of an earlier submission. The following is a list of the peer review reports and author responses from that submission.